# Integrated Pest Management of Wireworms (Coleoptera: Elateridae) and the Rhizosphere in Agroecosystems

**DOI:** 10.3390/insects13090769

**Published:** 2022-08-25

**Authors:** Atoosa Nikoukar, Arash Rashed

**Affiliations:** Southern Piedmont Research and Extension Center, Virginia Tech, Blackstone, VA 23824, USA

**Keywords:** wireworms, rhizosphere, IPM, click beetle, soil microbial communities

## Abstract

**Simple Summary:**

The name ‘wireworm’ refers to the subterranean larvae of click beetle (Coleoptera: Elateridae) species, of which several are serious pests of a wide range of crops. The limited effectiveness of the available insecticides, their wide host range, their long life cycle, and their cryptic subterranean habitat make wireworms a challenging pest to control. Integrated pest management (IPM) strategies have been recommended to reduce wireworm damage. Although IPM is generally considered to be an approach that is relatively more compatible with the environment and non-target organisms, the implementation of some of the tactics of managing subterranean wireworms is expected to induce stress in the rhizosphere and the established ecological interactions within, some of which could negatively impact various soil health parameters and, subsequently, plant growth. In this paper, we highlight some of the IPM tactics against wireworms and their effects on the rhizosphere and soil microbiome. Awareness of the potential impacts of IPM approaches to the management of subterranean pests will help professionals to develop and implement IPM strategies that minimize disturbance in the rhizosphere and support agroecosystem sustainability.

**Abstract:**

The rhizosphere is where plant roots, physical soil, and subterranean organisms interact to contribute to soil fertility and plant growth. In agroecosystems, the nature of the ecological interactions within the rhizosphere is highly dynamic due to constant disruptions from agricultural practices. The concept of integrated pest management (IPM) was developed in order to promote an approach which is complementary to the environment and non-target organisms, including natural enemies, by reducing the sole reliance on synthetic pesticides to control pests. However, some of the implemented integrated cultural and biological control practices may impact the rhizosphere, especially when targeting subterranean pests. Wireworms, the larval stage of click beetles (Coleoptera: Elateridae), are generalist herbivores and a voracious group of pests that are difficult to control. This paper introduces some existing challenges in wireworm IPM, and discusses the potential impacts of various control methods on the rhizosphere. The awareness of the potential implications of different pest management approaches on the rhizosphere will assist in decision-making and the selection of the control tactics with the least long-term adverse effects on the rhizosphere.

## 1. Introduction

Subterranean wireworms are the larvae of click beetles (Coleoptera: Elateridae; Figure 1A–C), and are one of the most challenging pests to manage in agroecosystems. The adult beetles emerge in spring and early summer, between April and June in the Pacific Northwest (PNW) and the Intermountain West regions of the US, to mate and lay eggs. After egg hatching, the emerged larvae can live in the soil for multiple years, where they complete seven to nine instar larvae within two to four years [1,2]. Wireworms are generalist herbivores [2], and they can cause damage to almost any crop in the rotation because of their multiyear larval stage and broad host range. The damage from wireworms can be reflected in yield reduction and quality loss (Figure 2A–C). The stand thinning resulting from their feeding on seed, emerging sprouts, and young seedlings allows for the establishment of weeds in the wireworm-infested areas [2]. At the field scale, the damage is often patchy, corresponding to the patchiness of wireworm distribution within fields, likely due to the adults’ oviposition site preference [3]. The duration of the larval stage can be influenced by food availability and environmental conditions; wireworms can survive extended periods without live vegetation, and nonliving organic matter constitutes a negligible portion of their diets [4,5,6].

Prior to the early 2000s, persistent broad-spectrum organochlorine insecticides were used to keep wireworm populations in check [1,2,7,8]. After the de-registration of organochlorines in the USA, due to their negative impact on non-target organisms and environmental risks, and as the persistent residual activity started to fade away, wireworms re-emerged as a key pest in several regions [9,10]. Despite recent developments in the chemical control of wireworms, insecticides have often failed to adequately control the wireworm population in most crops, especially small grains [11].

Integrated pest management (IPM) is an ecologically based approach that promotes a biorational use of pesticides based on some decision-making guidelines [12,13,14,15]. Although there has been growing interest in the implementation of IPM against wireworms, their subterranean habitat, patchy distribution, vertical movement in the soil, long life-cycle, broad host range, and most importantly, limited location and the species-specific knowledge of their ecology and phenology have made them a difficult pest to control. Despite being a biorational strategy, the implementation of IPM tactics for the management of wireworms can potentially alter the rhizosphere and the ecological interactions within; these effects are often overlooked [16]. The rhizosphere is an interface where plant roots, soil, subterranean pests, and beneficial organisms interact [16]. The agroecosystem’s productivity is dependent on soil fertility and healthy plant development, both of which are influenced by the microbial composition and an array of ecological interactions in the rhizosphere [17,18]. The ecological interactions in the rhizosphere are constantly disrupted by changes in plant species (e.g., crop selection and rotation), the crop developmental stage [19,20,21], pest management [22,23], and other farming practices [16,24,25]. For example, pesticide applications in non-organic production systems are known to alter the chemical and biological soil properties [26,27,28] that are essential for both soil and crop health [27,29,30]. Shifts in the soil microbiome can also, directly and indirectly, influence pest survival and susceptibility to the implemented control practices. For example, some soil microorganisms are known to influence insect feeding [31] or contribute to insecticide detoxification [32].

Therefore, practices that can alter the ecological interactions in the rhizosphere are expected to influence both target (i.e., subterranean pests) and non-target organisms (i.e., beneficial organisms, plants). It is important to note that the ultimate goal of pest management is to maximize the productivity of the agroecosystem. Awareness of the potential impacts of various recommended control tactics on the rhizosphere will assist with the development of IPM strategies that can more effectively contribute to the resilience and sustainability of the agroecosystem. This brief review will first provide an overview of the IPM components associated with the decision-making process which are pertinent to wireworm control. We will then discuss the effectiveness of each recommended control practice in the management of wireworm populations and damage. Under each approach, we will highlight the potential impacts on the rhizosphere and soil microbiome. Finally, we will discuss the implications of those effects on agroecosystem productivity and resilience, and identify the existing gaps needing further investigations.

## 2. Implementation of IPM Principles to Control Wireworms

### 2.1. World Fauna and Species Identification

Our current understanding of the global distribution of click beetles is limited, and the existing information is based on studies conducted in specific regions and countries. Moreover, species composition can change over time at both the field and global scales. The appearance of some of the European *Agriotes* spp. in North America also suggests that human activities can contribute to click beetle dispersal [2]. Overall, there are about 10,000 species (400 genera) of click beetles worldwide [33], of which 921 species (91 genera) are described in North America [34]. Three hundred and sixty-nine species have been reported from Canada and Alaska [35], of which 30 species are of economic importance [36]. In the PNW, *Limonius* spp. is the most common pest species of click beetles devastating sugar beets, small grains, potatoes, and organic vegetables [37,38,39,40].

In the Holarctic regions of Europe, North Africa, the Middle East, and North Asia, there are about 100 economically important species of click beetles [1,41]. *Agriotes* spp.—specifically *A. lineatus*, *A. obscurus*, and *A. sputator*—are Europe’s most damaging wireworm species [42]. *Agriotes* spp. have also been reported in North America [43], damaging various crops such as corn, potato, strawberries, and organic vegetables [44]. *Agriotes* spp. often is not considered to be a major pest taxon in most of the USA, where *Limonius* spp., *Selatosomus* spp., *Conoderus* spp., *Melanotus* spp., and *Dalopius* spp. represent some of the most damaging genera [2,45].

Several identification keys are available to determine adult genera and/or species [46,47,48,49]. However, the identification of the damaging immature wireworms is required in order to inform the selection of adequate and timely control tactics. The limited development of some morphological characteristics in the early larval instars [46], the absence of distinct features to distinguish some species [2], and/or worn-off morphological traits in late instars make larval identification a challenging task. Despite these constraints, the recent advances in molecular techniques have helped clarify several ambiguities in species identification. The *Agriotes* species complex [3] is one example in which species identification based on the larval morphology is challenging, and the cytochrome oxidase I (COI) and 16 S regions of the mitochondrial DNA helped determine species, and even revealed several haplotypes [50]. Moreover, DNA barcoding was used successfully to identify cryptic species within *Hadromorphus glaucus* Germer and *Hypnoidus bicolor* Eschscholtz [51,52]. More recently, COI and 16 S rRNA sequences and nuclear genome-wide single nucleotide polymorphisms generated from restriction site-associated DNA identified cryptic species within the *L. californicus* and *L. infuscatus*, where potential genomic responses to broad-spectrum insecticides were also detected [53]. Determining the effects of agricultural practices on population genetic diversity and identifying ecological variables that predict the evolution of adaptations and dispersal patterns [54] based on molecular data are emerging areas of research, and long-term studies with large sample sizes are needed in order to allow for the precise characterization of such effects.

### 2.2. Monitoring

#### 2.2.1. Adult Monitoring

Sex pheromone lures have been used to estimate the abundance and distribution of the male click beetles [3]. Synthesized female sex pheromones are now available for a few species of click beetles, including several European species [55], and a few pestiferous species in North America, including *Cardiophorus tenebrosus* [56], *Limonius* spp. [57,58], and *Melanotus communis* [59]. 

In order to estimate the subterranean wireworm populations and species composition, the monitoring of the adult click beetles with the pheromone traps was initially sought [2]. However, establishing the relation between the number of collected adult males and wireworms is complex, and is reflected in inconsistencies across study outcomes. For example, although a field study on *Agriotes* spp. by Benefer et al. [36] detected no associations between the number of click beetles and wireworms, Furlan et al. [60] reported a significant relationship between male click beetle numbers, wireworm populations, and the extent of damage to maize in *Agriotes* spp. Furlan et al. [60] also demonstrated that monitoring click beetle populations two years before planting maize could be a practical risk assessment method; in the case of *A. ustulatus*, capturing > 1000 click beetles/trap two years before planting was associated with a more-than-12-times probability of >15% crop damage. Click beetle behavior and their movement [61,62], the range of pheromone attractiveness [63,64], trapping time and intensity [3], sex ratio, climate, and agricultural practices [65] are examples of the variables that can influence the precision of adult monitoring in the prediction of subterranean wireworm populations. Although the development of pheromone lures provides us with sensitive tools to monitor click beetles [66] and the opportunity to develop distribution maps across landscapes [41], more large-scale species-specific data are needed in order to solidify associations between adult pheromone traps and wireworm bait trap data (Section 2.2.2).

#### 2.2.2. Wireworm Monitoring

Soil core removal [67] and solar bait traps [39] are two of the most frequently used field sampling methods [1]. Soil core sampling can be time-consuming [67,68]. Because the wireworm distribution is often patchy [5], many core samples would be needed in order to accurately estimate the field infestation status. Solar bait traps, on the other hand, are designed to attract wireworms [1], and typically stay in the soil for several days. Germinating cereal seeds are commonly used in solar bait traps as effective baits [69,70,71], attracting wireworms to the CO_2_ and other volatiles released from the sprouting seed [72]. Covering bait traps with soil, plastic [73,74], and charcoal dust [73] helps to increase the temperature and trap the released CO_2_ and volatiles in order to maximize wireworm attraction from a distance. Soil temperature [75], moisture [5,76], texture, and other existing sources of CO_2_ [68] may influence the effectiveness of the bait traps.

The effectiveness of wireworm monitoring also depends on the species-specific phenology and distribution patterns within fields [4,5,40,77]. Wireworm activity is influenced by food availability, molting, and abiotic variables such as moisture and temperature [2,7]. Wireworms dive deeper into the soil profile when the environment is unfavorable and/or become inactive [41,78,79]. Monitoring efforts when the wireworms are inactive or not present in the topsoil may not yield reliable results. Early-season sampling and prophylactic seed treatment application may miss the wireworm species that are active later in the growing season. For example, seed treatment in the PNW is expected to protect plants against *L. infuscatus*, which is active early in the season, but may not be as effective against *L. californicus*, which remains active throughout the season [40]. Continuous monitoring during the growing season can help capture an accurate estimate of wireworm populations, species composition, and distribution.

Wireworm distribution within an infested field is often patchy, and multiple species of wireworms can be present within a field [2,36] (AN and AR, unpublished). Because wireworms have limited ability to disperse within the soil [80], their patchy distribution may be explained by the adult female preference for oviposition sites [3]. The adult click beetles are attracted to grasslands for oviposition [2,3,81], and higher wireworm populations can be found in pastures, cereal fields lacking proper rotation, and natural vegetations within and surrounding fields [82,83]. Wireworm distribution and the accuracy of field monitoring can also be driven by other environmental constraints such as soil characteristics and texture [3,84].

### 2.3. The Risk of Wireworm Damage

#### 2.3.1. Wireworm Economic Threshold

The number of wireworms is not the only variable determining the risk of damage; threshold assessments should also consider wireworm species and the crop of interest [85,86,87]. Cherry et al. [88] estimated that nine larvae of *Melanotus communis* in 25 soil samples collected from infested sugarcane fields could cause severe economic damage. In another study, Furlan [87] evaluated the damage of *Agriotes* spp. in corn, and showed that the effect is influenced by both the number and species of the wireworms. The damage by one larva of *A. brevis* was equal to damage caused by two *A. sordidus,* or five *A*. ustulatus. In potatoes, seed treatment is recommended if the number of collected *L. californicus* or *L. canus* per trap exceeds two larvae [89]. Moreover, different crops show different levels of susceptibility to wireworm damage; *Agriotes* spp. can cause 100% and 50% damage in sugar beet and corn, respectively, whereas the damage in soybean remains negligible [77].

#### 2.3.2. Soil Properties and the Risk of Wireworm Damage

Soil temperature, moisture, and texture are among the most important variables associated with wireworm damage [2,68,78,79,90,91,92,93].

In a study conducted in eastern Canada, wireworms started their activity when soil temperatures reached slightly above 1.5 °C, and their movement peaked when soil temperatures reached 12 °C; *Dalopius pallidus*, *Ctenicera lobata*, *A. mancus*, and *H. abbreviatus* were the five species in the study sites [94]. These reported temperatures are similar to those experienced by wireworms in the early spring when spring-seeded crops are sprouting. Soil temperatures between 8 and 14 °C and soil moistures between 30 and 32% are associated with a high risk of crop damage by *Agriotes* spp. [79]. The preferred soil moisture of wireworms is also influenced by the soil’s physical structure [79]; the soil texture and compaction influence fluctuations in moisture and temperature, and affect the ability of wireworms to relocate and survive [95,96,97,98]. Soil porosity can also affect the diffusion of carbon dioxide (CO_2_) and other volatile compounds released from the germinating seeds and roots, which are cues that wireworms use to navigate food sources [41,81,99]. A greenhouse study showed that the frequency of damage by *L. californicus* to small grains increases in sand-dominated media [100]. Hermann et al. [95] also reported a positive association between sandy soil and potato tuber damage by *Agriotes* sp. Rapid changes in moisture levels in sandy soil have been proposed as a factor triggering wireworms to burrow into succulent tuber tissue in search of moisture.

Other soil variables that are studied in relation to the risk of wireworm damage are the pH and the organic matter content [101]. However, it is also important to note that the wireworm response to soil pH and soil organic matter can be species-dependent. For example, *Agriotes* and *Melanotus* prefer acidic soils [83], whereas *Limonius* spp. can be active in both alkaline and acidic soils [83]. Soil organic matter is not a major food source for wireworms [6,102], and a long-term survey has suggested an increased risk of *Agriotes* spp. presence and damage in soils with a high organic matter content [103]. The contribution of soil organic matter to retaining moisture is likely to make conditions favorable for wireworm survival. While the influence of environmental variables on wireworms is often studied (or presented) in isolation, wireworms (and all living organisms) are always exposed to a combination of variables. An understanding of the species-specific wireworm response to combinations of interacting abiotic factors in the rhizosphere is expected to help with the development of relatively more effective control methods (see Section 3.1.4, for an example).

#### 2.3.3. Landscape and Field History and the Risk of Wireworm Damage

The landscape surrounding farmlands and cropping history can also predict the risk of wireworm infestation [95]. Specifically, grassland’s (including small grains with no rotation) presence and duration are strong predictors of the likelihood of wireworm presence [97]. 

Anecdotal observations are also suggestive of increased wireworm damage on hillsides and slopes (Rashed, personal observations), which supports the trend observed by Parker and Seeney [97], who reported a higher frequency of wireworm infestation in south-facing fields. As mentioned earlier, meadows, natural grass patches (including those dedicated to the ‘Conservation Reserve Program’), and grassy and weedy field margins considerably increase the wireworm damage risk because they attract adult females and provide a favorable place for oviposition and larval development [7,68,95,103,104].

Poggi et al. [105] provided a comprehensive list of factors, including agricultural practices studied for their impacts on the damage risk of wireworms. However, often overlooked are the potential implications of the implemented control practices on the rhizosphere and the ecological interactions involving subterranean micro- and macro-organisms. 

## 3. Wireworm Control Tactics and Impacts on the Rhizosphere

### 3.1. Cultural Practices

#### 3.1.1. Trap Crop and Intercropping

Because wireworms are limited in their mobility, it may appear that trap cropping would not be a practical approach to reducing crop damage. However, placing attractive trap crops within the main crop has been shown to protect the main crop [43,106]. Intercropping with wheat as the trap crop helped reduce wireworm damage in strawberries [107] and maize [108]. The effectiveness of an attract-and-kill strategy, using insecticide-treated wheat as a trap crop, has also been tested and proven successful in potatoes [109].

Intercropping can negatively impact pests by interfering with their ability to locate host plants and/or by providing an environment which is suitable for the pest’s natural enemies [16,110]. There is also evidence that diversity in trap crops (i.e., using seed mixes) can increase wireworm attraction away from the main crop. Staudacher et al. [111] compared the effectiveness of wheat and a mixture of the wheat, bean, lupine, white mustard, buckwheat, and ryegrass as trap crops in maize; greater protection of maize seedlings was achieved with the mixed-species trap crop.

Intercropping can also positively impact the agroecosystem and soil microbial communities, enhancing plant productivity [112,113,114,115] . However, the microbial community composition and diversity in the rhizosphere can also shift in response to the soil type, plant root exudates, and the interaction between plant species, i.e., nutrient availability and uptake efficiency [112,116,117,118,119,120]. Therefore, selecting suitable plant species for trap cropping is essential in order to sustain soil health, support vigorous plant development, and protect the main crop.

The associations between the nitrogen (N) fixing soil bacteria in legumes make pulse crops suitable candidates for intercropping [115,119,121]. The root tissues of legumes are a source of high organic phosphorus (P) [122], and through the nitrogen-fixing process they increase N in the soil [123]. Intercropping with legumes reduces the need for fertilizers; therefore, it minimizes nitrate and nitrite pollution in the soil [115]. In cereal production, using legumes as intercrops enhances yield and increases microbial diversity and biomass in the rhizosphere [112,113,117,118,119,124,125]. Legumes, such as pea and lentil, are also effective trap crops, attracting wireworms away from the main wheat crop and, subsequently, reducing damage [126,127,128]. Peas planted in potato fields have also been more attractive than wheat and oilseed to wireworms in field trials [126].

Although intercropping with legumes can potentially reduce wireworm damage in some crops, such as small grains and potatoes, the contribution of this approach to N fixation and improved microbial activity could also, in turn, create an environment in the rhizosphere that supports wireworm development and survival through improved N metabolism and availability [129,130] and/or by suppressing the efficacy of entomopathogens [131,132]. When it is abundant, wireworm activity can also increase the microbiome density, functional diversity, and mineralization rate [129]. For example, in higher N and carbon contents, wireworms can shift the soil bacterial composition to increase the abundance of *Azotobacter* bacteria, which contribute to nitrogen fixation in the soil [129,130]. 

#### 3.1.2. Crop Rotation

Crop rotation can reduce pest incidence (i.e., pathogens, weeds, and arthropods) [133,134] and contribute to N recovery, nutrient availability, and an increased rate of mineralization [135,136]. However, because wireworms are polyphagous, they can feed on various crops in rotations. Therefore, finding a proper diversified rotation to minimize wireworm damage is particularly challenging. The continuous rotation of small grains with susceptible crops such as potato and sugar beet is not expected to reduce wireworm damage [7,82,93,104,137]. However, rotations with relatively less-susceptible crops such as mustard, soybean, sorghum, cabbage, French marigold, clover, and flax could help to reduce wireworm damage when followed with more susceptible target crops [138]. Long rotations with alfalfa can negatively impact the wireworm population [139]; this effect is attributed to the soil drying and compaction in the rhizosphere due to the high root density developed over the years [140].

A diverse crop rotation plan is expected to enhance the soil microbiome richness [109,136], diversity [24,136,141,142], and crop yield [136,143]. This is because crops differ in the carbon resources that they contribute to the rhizosphere, either through the release of species-specific root exudates or the plant residue that they leave behind in the soil [144,145,146,147], continuously selecting for the microbiome that can utilize the available resources. As discussed earlier, an increase in the diversity and activity of the microbial community in the rhizosphere may not directly negatively impact wireworm populations. Still, it could indirectly reduce damage by supporting vigorous plant growth (i.e., the subsequent crop), which may withstand and overcome feeding by the pest. Moreover, some of the N-fixing crops, such as pea, bean, lentil, and alfalfa, which are relatively more tolerant of wireworm damage, may be potential candidates for rotation with susceptible crops [138]. The selection of a crop rotation plan that fits the regional cropping system and the soil’s physicochemical characteristics is of particular importance. In relation to this, Nettles [148] states: “Several rotations have been suggested by research workers on the wireworm problem, but it should be borne in mind that no one rotation plan will suite all farmers.” In the PNW, peas and lentils are among the most common crops planted in rotation with cereals in rainfed production systems, which are relatively more tolerant of wireworm damage. Promoting diverse long-term rotation with legumes benefits the soil agroecosystem and may help to keep wireworm damage in check.

#### 3.1.3. Tillage

Overall, conventional tillage can negatively impact the wireworm population and reduce the risk of damage to several economically important crops [82,99,149,150]. The effectiveness of tillage as a control method depends on the wireworm’s seasonal activity and life cycle; tillage can be a practical approach when larvae are active in the upper layer of the soil surface [5,149]. Disturbing the topsoil with tillage can expose the click beetle eggs and the newly hatched wireworms to predators, heat, and desiccation [5,77,151]. However, tillage can also stress the rhizosphere by altering the physical structure, aeration, moisture, organic matter decomposition and nutrients, and ecological interactions in the subterranean community that support root development and plant growth [152]. As such, in recent years, conservation agriculture has been promoting minimum soil disturbance (i.e., reduced or no tillage) to improve and conserve soil quality, preserve water, and minimize soil erosion [153,154,155]. The no-tillage practice, with the coverage of plant residue on the soil surface, improves soil microbial diversity and soil organic matter compared to the conventional tillage system [25,141,156], and contributes to the resilience of the agroecosystem [152]. The relatively more stable rhizosphere under no-tillage is expected to provide a suitable environment for wireworms to survive, and has been proposed as an underlying cause for their resurgence as a key pest (e.g., [7]). However, a recent study concluded that in the maize production system, no-tillage neither increased wireworm populations (primarily *A. sordidus*) nor the rate of damage [157].

While no-tillage and reduced tillage can contribute to soil health and the resilience of the agroecosystem, their impact on wireworm populations and damage require further studies that are species-, crop- and location-specific [41].

#### 3.1.4. Soil Flooding and Drying

Due to the importance of soil moisture in wireworm survival [37,40], water manipulation has been recommended as a possible control approach for wireworms [98,107,158,159]. The effect of soil drying is species-specific; withholding irrigation soon after oviposition to dry topsoil reduced *Agriotes* spp. [5,68], *L. californicus*, and *L. canus* [78] numbers. However, this approach was proven to be ineffective against the dryland species *Selatosomus pruininus*, known as the Great Basin wireworm [159]. 

The efficacy of flooding as a control approach against wireworms is both species- and temperature-dependent. Flooding can effectively control *L. californicus* when the soil temperature exceeds 21 °C [158]. Similarly, in flooded soil, *A. obscurus* and *A. lineatus* died more rapidly in 20 °C—an effect which was more pronounced in soils with high salinity [98]. Unlike the *Agriotes* and *Limonius* species complexes, the prairie grain wireworm, *Selatosomus destructor* (AKA. *Ctenicera destructor*), can feed and molt while submerged in water for up to 6 weeks [160]. 

Although water manipulation may help to reduce some species of wireworms, both flooding and drought are known to influence biotic and abiotic interactions in the rhizosphere. For example, soil microbial biomass and the bacteria:fungi ratio of the microbiome have been reported to decrease in the flooded, anaerobic environment, especially in 10–20 cm soil depths [161]. A reduction in soil microorganisms may provide the entomopathogenic fungi and nematodes with a competitive edge, improving their efficacy against wireworms. There are studies demonstrating the antagonistic effects of indigenous soil microbes on the applied entomopathogenic agents [162]. The entomopathogenic nematodes *Heterorhabditis bacteriophora* and *Steinernema feltiae* have been shown to be more effective against mealworms (*Tenebrio molitor*) in sterilized compared to non-sterilized soil [131]. In another study, Mazzola [163] suppressed beneficial soil bacteria using soil fumigation in order to apply biological control agents against apple soil-borne pathogens more effectively.

Prolonged periods of drought before planting can also have adverse and long-lasting effects on soil bacterial communities; the addition of N fertilizer failed to improve soil fertility and recover the soil bacterial community [164]. The potential negative impacts of soil drying and flooding as a wireworm control method on soil health parameters need to be further studied and quantified.

### 3.2. Host Plant Resistance and Tolerance

Because of the wireworms’ subterranean habitat and often-unknown species-specific seasonality, planting resistant and tolerant host plants can be a reliable approach to reducing the damage risk. Potato, corn, and sweet potato are the better-studied crops for variations in susceptibility to wireworm damage. Johnson et al. [165] studied the susceptibility of 12 potato varieties to wireworm feeding. There, the glycoalkaloid concentrations were suggested as one of the mechanisms contributing to the observed differences in susceptibility to *Agriotes* spp.

Differences in susceptibility to *A. sordidus* were also reported among corn varieties [166], where the less susceptible corn cultivars released higher concentrations and a more diverse blend of volatile compounds such as hexanal, heptanal, and 2,3-octenanedione than the susceptible genotypes [166]. Hexanal and several other aldehydes have also been detected in barley roots [167], which are a more tolerant crop to wireworm feeding than wheat [100,168]. Oats show resistance to wireworms [168], likely because of their vigorous root development and the presence of defensive steroidal and triterpenoid saponins [169]. 

Plant roots influence soil microbial composition and activity because they (the roots) shed cells and release various species-specific metabolites into the rhizosphere [170,171]. At the same time, these interactions may develop into a plant-specific microbiome in the rhizosphere (e.g., [116,172,173]). The persistence of such effects needs to be studied in agroecosystems where crop rotations are common. The potential impact of plant defenses on non-pathogenic microorganisms in the rhizosphere is an area that needs further investigation (see [174] for a review). 

### 3.3. Biological Control

#### 3.3.1. Predators

Besides general vertebrate predators such as birds [175], moles and amphibians [2], arthropods including carabid beetles, rove beetles (Coleoptera: Staphylinidae) [176,177] and the stiletto fly *Thereva nobilitata* (Diptera: Therevidae) larvae [178] have been reported to prey upon wireworms. It is conceivable that, like wireworms, the soil-dwelling predatory beetles and their subterranean larvae are susceptible to some of the agricultural practices that disturb the rhizosphere. Although the impact of these predatory insects on wireworms is more limited than the entomopathogenic organisms, additional studies are needed in order to quantify their impact as biological control agents. Several species of parasitoid wasps from the families Proctotupidae, Bethylidae, and Ichneumonidae, and flies from the family Tachinidae have also been reported to attack several species of wireworms, but again the extent of parasitism in the field appears to be limited [2]. 

#### 3.3.2. Entomopathogenic Bacteria

Among the resident bacteria in the rhizosphere, *Pseudomonas aeruginosa* has been reported to infect wireworms during the susceptible molting process [179]. Although the application of entomopathogenic bacteria to control wireworms has not been thoroughly investigated, the modification of the endosymbiotic bacteria associated with wireworms as a biological control method has been the subject of a few studies [180,181,182]. Lacey et al. [183] isolated *Rahnella aquatilis*, a common bacterium in the rhizosphere in the potato agroecosystem, from *L. canus*. The genetically modified *R. aquatilis* expressed wireworm-active toxins and successfully protected seed potatoes from wireworm damage [182]. *Rickettsiella agriotidis* isolated from *Agriotes* spp. [181], and *Arthrobacter gandavensis, Bacillus thuringiensis,* and *Pseudomonas plecoglossicida* isolated from *A. lineatus* are a few more examples of bacteria [180] with the potential to be developed into biological control agents against wireworms. 

#### 3.3.3. Entomopathogenic Nematodes

*Steinernema* and *Heterorhabditis* (Nematoda: Steinernematidae, Heterorhabditidae), along with their endosymbiont bacteria *Xenorhabdus* spp. and *Photorhabdus* spp., respectively, are the two main genera of entomopathogenic nematodes (hereafter, EPN) [183]. Although the EPN are expected to perform well in protected and moist habitats, a few challenges exist with regard to wireworms [184]. Their resilience, physical barriers, and behavioral responses make wireworms a difficult host to infect [185]. In the laboratory, *Steinernema carpocapsae, S. riobrave* and, *S. glaseri* reduced damage caused by the sugar beet wireworm, *L. californicus* [186,187]. *Heterorhabditis bacteriophora*, *S. carpocapsae,* and *S. feltiae* were used to manage *A. lineatus* [188,189], *A. obscrus**,* and *A. sputator* [189]; the three entomopathogenic nematodes were effective against *Agriotes* spp. However, *S. feltiae* has not been effective in controlling *S. aeneus* [190].

Differences in the efficacy of EPN species may be explained by differences in their foraging behavior [183,191] and/or species-specific differences in wireworm ecology [40]. The efficacy of EPN is also influenced by biotic and abiotic environmental variables [192,193,194]. Therefore, naturally occurring EPN adapted to their soil habitat and endemic hosts might be better than commercial nematode species for the management of local wireworms [184,195,196,197]. More studies are needed in order to evaluate the efficacy of naturally occurring nematodes against wireworms. 

#### 3.3.4. Entomopathogenic Fungi (EPF)

Overall, EPF are a promising tool in the management of wireworms. *Metarhizium anisopliae* (Clavicipitaceae) and *Beauveria bassiana* (Cordicipitaceae) recovered from the larvae and adults of *Hypolithus bicolor* caused significant mortality in *H. bicolor* and *S. aeripennis destructor* [198]. The *B. bassiana* isolated from the genera *Agriotes*, *Conoderus*, and *Hypnoidus* [2] successfully reduced *Agriotes* spp. damage in potatoes [199]. The in-furrow application of *B. bassiana* also reduced the number of *Limonius* spp. collected in bait traps in organic fields (unpublished data). However, in a study by Kölliker and colleagues [200], comparing efficacies of *M. anisopliae* (isolate ART-2825) and the commercial *B. bassiana* against *Agriotes* spp., the two EPF species differed significantly in their efficacy against the wireworm, with *B. bassiana* being ineffective. This finding supported Ansari et al. [189], who determined *B. bassiana* to be non-pathogenic against *A. lineatus*. Although the efficacy of *M. anisopliae* reached as high as 97% against *A. obscurus*, it was considerably less virulent against *A. lineatus* and *A. sputator* [200].

Ansari et al. [189] also demonstrated that *M. anisopliae* can cause mortality in *A. lineatus*; however, the mortality rate varied between 10 and 100%, depending on the EPF isolate. Later, Reddy and colleagues [66] reported a reduced number of *L. californicus* and *H. bicolor* in experimental plots treated with *M. brunneum* (F52)*, M. robertsii* (DWR346), and *B. bassiana* (GHA), with no difference being detected in the efficacy of the EPF species. Field-collected EPF isolates have also shown promise in the reduction of wireworms. For example, *M. anisopliae* isolated from field-collected *A. obscurus* was effective against that species in field trials when applied at the high rate of 4 million conidia/cm3 [201]. 

Application methods that facilitate contact between the biological control agent and wireworms can result in successful infection in the rhizosphere where wireworms are most active; in-furrow and soil drench applications of EPF were more effective than seed coating applications [66,202,203]. However, *M. brunneum* (1154) still successfully reduced the wireworm population and crop damage when applied as a seed coating [204].

Biotic and abiotic environmental variables in the rhizosphere can determine the efficacy of entomopathogens. The soil temperature [205,206,207], moisture [201,208], and texture [192,209]; the duration of exposure; the pathogen concentrations [206]; the wireworm location at the time of application [202]; and interactions with the soil microbiome [131,132,162,210] are examples of factors that can predict the efficacy of the entomopathogenic organisms against wireworms. 

Fluctuations in soil moisture and temperature are affected by the soil porosity and compaction, which are traits that also influence the ability of wireworms and entomopathogens to relocate and survive [95,96,97,98,192]. For example, a porous sandy texture can facilitate EPN locomotion and foraging success [192,193], but it may also increase the rate of damage by wireworms [100]. High soil organic matter is expected to support EPF, likely because soils with high organic matter help with moisture retention [211]. Therefore, selecting entomopathogens which are compatible with environmental and field conditions is critical if the biological control approach is being implemented alone or as a component of IPM against wireworms [66,206]. 

Soil temperatures above 18 °C are optimal for *M. anisopliae* to infect wireworms successfully. As wireworms move deeper into the soil in order to evade increasing temperatures and the dry topsoil, fungal conidia may not be able to reach the wireworms [202] effectively; the fatal exposure time for wireworms at 18 °C is estimated to be at least 48 h [206]. Moreover, soil moisture below 6–18% [201] negatively impacts the viability of the fungal conidia, further reducing the efficacy of the EPF.

The optimal temperature to reach higher infectivity differs among EPN species. For example, the pathogenicity of *S. carpocapsae* [194] and *H. bacteriophora* [212] at temperatures ranging from 5–25 °C is greater than 35 °C, whereas *S. glaseri* shows higher pathogenicity at temperatures ranging between 15 and 35 °C [194]. Understanding the species-specific ecology of wireworms and identifying the most effective species and strain of entomopathogens for the environmental conditions are essential variables that can determine the success of the biological control.

The subterranean entomopathogens continuously interact with the microbial organisms in the rhizosphere. These interactions may affect the virulence of the entomopathogens, as well as their ability to survive, move, and locate the host. Susurluk [131] and Shah [132] showed that the efficacy of *H. bacteriophora*, *S. feltiae*, and *M. anisopliae* in sterilized soil is significantly higher than that in non-sterile soil due to microbial suppression in sterilized soil. As mentioned earlier, cultural practices such as trap cropping/intercropping [112,113,114,115] and no-tillage [25,141,156] promote the microbial biomass, diversity, and activity in the rhizosphere, which in turn could interfere with the efficacy of the entomopathogens against subterranean pests. Future studies are warranted to evaluate these potential trade-offs in agroecosystems. Moreover, while wireworms may directly benefit by contributing to N fixation in the soil and enhancing microbial activity and biomass [129], they could also benefit indirectly as the effectiveness of the entomopathogenic organisms is reduced. However, it is also important to note that some entomopathogens, such as *Metarhizium,* are also known to colonize plant roots, promoting growth, drought resistance, and nitrogen acquisition by the host in the rhizosphere [213,214,215].

One of the less-studied aspects of wireworm management is the role that the associated endosymbionts may play in the determination of wireworm susceptibility to entomopathogenic organisms. Kabaluk et al. [216] reported that the four endosymbiont bacteria *Pantoea agglomerans, Pandoraea pnommenusa, Nacardia pseudovaccinii*, and *Mycobacterium frederiksbergense*, associated with wireworm species *A. obscurus* and *A. lineatus*, can suppress infection by efficacious *M. brunneum*. 

Although several studies examined the efficacy of various strains of entomopathogenic organisms in isolation, in the rhizosphere wireworms are often exposed to multiple natural enemies simultaneously [217]. A few studies examined the efficacy of a combination of EPF and EPN against wireworms, which yielded inconsistent results [217,218,219,220]. The co-application of EPN and EPF has been reported to result in synergistic [218], additive [219], or even antagonistic [220] outcomes. The time and order of the applications, the developmental stage of the pest [221], the species and strain of the entomopathogens [222], and soil conditions [223] are examples of the variables that may influence the outcome of co-applications of the entomopathogens.

Entomopathogenic organisms have also been used in combination with insecticides. In the laboratory, combining *M. anisopliae* with the insecticide spinosad, a natural insecticide produced by the soil bacterium *Saccharopolyspora spinosa*, increased mortality in *A. obscurus* and *A. lineatus* [224]. In the field, corn seed coated with *M. anisopliae* and either spinosad or clothianidin (neonicotinoid) was ineffective against *A. obscurus* [213]. However, the application of *M. anisopliae* alone significantly increased the yield [213]. Mixed applications of the bioinsecticides spinosad, azadirachtin, pyrethrin, *M. brunneum* F52, *B. bassiana* ANT-03, and *B. bassina* GHA with each other and in combinations with the neonicotinoid imidacloprid have also been evaluated against the three wireworm species *L. californicus, H. bicolor, and Aeolus mellillus* in spring wheat [207]. The combined applications of imidacloprid + *M. brunneum* and *M. brunneum* + spinosad protected wheat seedlings from the wireworms *L. californicus* and *H. bicolor*. However, the wireworm population treated with *M. brunneum* + spinosad was significantly higher compared to the control [207].

The attract-and-kill approach, based on wireworm responses to attractants such as CO_2_ and other plant volatiles has also been used to increase contact between biocontrol agents and wireworms [109]. Brandle et al. [209] used *M. brunneum* in combination with capsules of baker’s yeast as an artificial source of CO_2_ to attract *Agriotes* spp. in organic potato fields, and reported a significant reduction in crop damage. Millet grain has also been used as an attractant to draw wireworms to *B. bassiana* and *M. brunneum* in spring wheat [208]. Kabaluk et al. [201] used germinating wheat seeds coated with *M. anisopliae* as a bait trap against *A. obscurus* in the field. Finally, in the laboratory, alginate beads loaded with a combination of potato extract (as an attractant) and EPN (either *S. carpocapsae* or *H. bacteriophora*) were effective in attracting and killing *A. obscurus* [225].

Using natural enemies to reduce wireworm populations can be costly, especially in large-scale production systems. Therefore, if the natural enemies can persist in the rhizosphere and remain viable for extended periods, their use as a control tactic could be justified from an economic standpoint. In agroecosystems, the rhizosphere is disturbed frequently by various cultural practices, which could influence the populations of natural enemies. For example, entomopathogenic fungi densities are higher in habitats with less soil disturbance, such as grasslands, than cultivated fields [226]. Location-specific studies are needed in order to evaluate further the persistency of the applied biological control agents and the naturally occurring entomopathogens in relation to cultural practices in various agroecosystems.

### 3.4. Cover Crops and Plant-Derived Biocides as Green Manure

The integration of cover crops into the crop rotation plans can contribute to functional biodiversity, improved pest management [227,228], and nutrient availability [229,230,231] in agroecosystems. There are several species of plants from the family Brassicaceae that contain biocidal glucosinolates, which are used to manage arthropod, pathogen, and weed pests in agricultural systems [110,232,233,234,235]. When hydrolyzed, the biologically inactive glucosinolates produce species-specific and active biocidal compounds [236]. The major glucosinolate in the yellow mustard *Sinapis alba* is sinalbin, which produces ionic thiocyanate SCN-, a phytotoxic biocidal product [237]. SCN- is ineffective against wireworms [238,239], and is known primarily for its herbicidal effects [240,241]. In the brown mustard *Brassica juncea*, sinigrin hydrolyzes to produce the volatile 2-propenyl isothiocyanate, a product with proven efficacy against insect pests such as wireworms [242]. The glucosinolates in *B. carinata*, *B. oleracea,* and *B. nigra* are also known for their insecticidal effects on soil-borne insects [243,244,245]. Mustard products with a concentrated glucosinolate content have also been developed and tested against wireworms. For example, defatted *B. carinata* seed meal caused higher mortality in *Agriotes* spp. compared to soil-incorporated *B. juncea* plant tissue [238,245]. Although mustard seed meal can have higher efficacy against wireworms, its application in the field has been challenging due to its bulkiness. In order to address this limitation, a highly concentrated seed meal extract has been developed from *B. juncea* seed meal [236], which successfully suppressed the reproduction of the plant pathogenic cyst nematode, *Globodera* spp. [246]. This concentrated seed meal extract also effectively reduced the *L. californicus* population in the laboratory (unpublished data). Although these newly developed concentrated extracts are relatively easier to apply in the field, their effectiveness and potential impact on the activities of soil microorganisms have yet to be evaluated in the field.

Plant biofumigants are expected to degrade rapidly in the soil [243,245,247]. Hence, green manuring the mustard cover crop is thought to benefit the soil by contributing nutrients and organic matter and, subsequently, improving yield [235]. However, Hansen et al. [248] reported a significant reduction in fungal and mycorrhizal community and microbial biomass in general in the long-term wheat/canola (*B. napus* L.) rotation compared to the spring wheat/winter wheat rotation. Although the canola contains considerably lower concentrations of glucosinolates than either *B. juncea* or *S. alba*, the continuous release of glucosinolates was proposed as a reason for the decline in the microbial community and, subsequently, the wheat yield [248,249]. The effects of glucosinolates and their bioactive derivatives are not limited to soil microorganisms and insect pests; beneficial organisms such as entomopathogenic nematodes and fungi are also negatively impacted [110], which may interfere with the effectiveness of wireworm IPM. However, as discussed earlier, the post-fumigation soil application of entomopathogens has been reported to enhance the efficacy of biological control agents because it reduces the competition with the microbial communities in the rhizosphere [163].

The effects of mustard and its products on soil health in various cropping systems need thorough investigations. The effective dosage based on the infestation rate, the degradation parameters, the homogenous spread of seed meal products across the field, effective incorporation into the soil, the optimal soil temperature and moisture, and most importantly, the timely incorporation of green manure with respect to species-specific wireworm ecology, are a few factors that can predict the probability of success in wireworm management using biofumigants [41,77,238].

### 3.5. Insecticides

Organochlorines are a group of insecticides with persistent environmental effects [250]; they were used for decades to control agricultural pests, including wireworms [1,3]. This group of insecticides is now banned in the USA and Canada due to environmental and human health risks. Organophosphates (chlorpyrifos, fonofos) and carbamates are two other groups of relatively less persistent insecticides which are used to control soil-dwelling pests such as wireworms, and are again mostly deregistered in the USA [11,251]. Phenylpyrazole, pyrethroids, and neonicotinoids are now the most common groups of insecticides used to control wireworms, offering various degrees of efficacy against this pest [9,11,252,253].

Some pyrethroids, such as tefluthrin [9,11,252], and neonicotinoids, such as thiamethoxam, imidacloprid, and clothianidin [9,10], are known to act as feeding deterrents, and are used as seed treatments to reduce wireworm damage. Although these insecticides are not effective in lowering wireworm populations [252], they are expected to provide crop protection during the vulnerable stages of plant development [254]. The mixed application of insecticides can also improve their efficacy against wireworms [10,252,254]. In the laboratory, combinations of thiamethoxam + fipronil [254,255], thiamethoxam + chlorpyrifos [254], and thiamethoxam + pyrethroid [253] showed higher efficacy against different wireworm species than individual applications of those insecticides.

The soil microbiome plays a critical role in degrading pesticides and removing them from the rhizosphere [251]. Applying insecticides against subterranean pests could alter the chemical and biological properties of the soil. Neonicotinoids, which are one of the most commonly used seed treatments in small grains, are known to alter the structure, genetic diversity, and bioactivity of the soil microbial communities, at least in the short-term [32,256,257,258]. Applying a higher dosage of the neonicotinoid imidacloprid can result in long-term changes in the soil’s microbial composition and metabolic activity [256]. Another neonicotinoid, thiamethoxam, has been reported to temporarily reduce the bacterial abundance of some of plant growth-promoting bacteria, such as *Actinobacteria*. In contrast, pollutant-degrading bacteria (*Firmicutes*) are increased in the rhizosphere following insecticide applications [32]. The soil microbiome influences the composition of endosymbionts associated with herbivorous insects [21,23,31]. Therefore, changes in the soil microbial community following insecticide applications could potentially impact the endosymbionts associated with wireworms. Understanding the functional role of various endosymbionts in the life history traits of wireworms is a developing area of research. Symbiotic microorganisms are known to promote insect resistance to biological control agents like *B. thuringiensis* in gypsy moth [259], *M. brunneum* in wireworms [216], and parasitoid wasps in aphids [259]. In the context of resistance to insecticides, *Burkholderia* is an insecticide-degrading group of bacteria present in the soil, and can also be found in the midgut of the stink bug, *Riptortus pedestrist* [22,260]. There, the soil application of the organophosphate fenitrothion resulted in a shift in the composition of the soil microbiome, increasing insecticide-degrading bacteria, which can be ingested by the herbivores, promoting resistance to insecticides [22].

Despite these reported impacts on the soil microbiome after insecticide applications, the rhizosphere microbiome is expected to reach a stable status within months [257]; soil organic matter, pH, and texture, have direct implications for the leaching, adsorption, and desorption of the applied insecticides [261,262].

The metadiamide broflanilide is the most recent insecticide developed to reduce wireworms in small grains. Although this insecticide has been very effective in knocking down wireworm numbers, little is known about its impact on the rhizosphere and non-target organisms. Nonetheless, our current knowledge of pesticide impacts on the agroecosystem-specific rhizosphere microbiome with respect to soil physiochemical properties is limited [28,263], and additional studies are needed in order to determine whether the adverse effects of pesticides on the non-target organisms in the rhizosphere outweigh the benefits of reduced wireworm damage.

## 4. Conclusions and Future Directions

The abiotic components of the rhizosphere, e.g., the soil texture, mineral composition, organic matter, and moisture, can be directly affected by the cultural practices recommended for wireworm management, such as tillage, soil flooding and drying, and long-term crop rotations (e.g., alfalfa). However, several soil health parameters—such as microbial biomass, diversity, and activity—can also be negatively impacted by such practices.

Intercropping, planting less susceptible crops, and incorporating biofumigants and cover crops into rotation plans to reduce wireworm populations can cause fluctuations in the microbial community structure and its functional role. The soil microbiome and function can also be impacted by the plant species, variety, and physiological stage of plant development [264,265,266,267]. Moreover, although insecticides may temporarily reduce wireworm losses, they can negatively impact the microorganisms that support healthy plant development. Alternatively, or in addition, the increase in pesticide-degrading bacteria following insecticide applications may increase the rate of their ingestion by herbivores, thereby promoting resistance to the chemistry, which could become heritable in a population [22]. This is a critical area of investigation with regard to the development of sustainable pest management strategies, which to our knowledge have yet to be studied in wireworms.

This brief review was intended to bring attention to the fact that the ultimate goals of IPM are to minimize losses to insect pests and to contribute to the sustainability of the production systems. However, many of the recommended management practices to reduce wireworm populations and damage could potentially negatively impact variables that contribute to soil and crop health. While studies are needed to examine the tradeoffs among various practices, recognizing some of these antagonistic effects may also help to improve the effectiveness of integrated control strategies. For example, the efficacy of entomopathogens against wireworms may be improved if they are applied after biofumigation or soil flooding or drying, which are practices that are expected to temporarily suppress microbial populations in the topsoil.

Our knowledge of the endosymbionts associated with wireworms is limited to a few studies on a very limited number of species [130,180,181,182,216,268]. Future studies are warranted in order to further learn about the species-specific wireworm endosymbionts in relation to the location-specific soil microbiome and understand their functional roles with respect to wireworm development and survival (e.g., resistance to stress).

The understanding of the wider impact of IPM practices on the sustainability of agroecosystems is a developing and continuously evolving area of research, which requires a transdisciplinary framework based on an in-depth interdisciplinary approach to future studies of complex ecological interactions.

## Figures and Tables

**Figure 1 insects-13-00769-f001:**
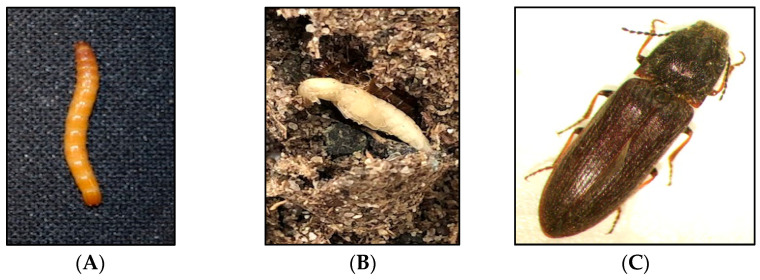
Different developmental stages of the click beetle, *Limonius californicus*: (**A**) late instar larva, (**B**) pupa, and (**C**) adult.

**Figure 2 insects-13-00769-f002:**
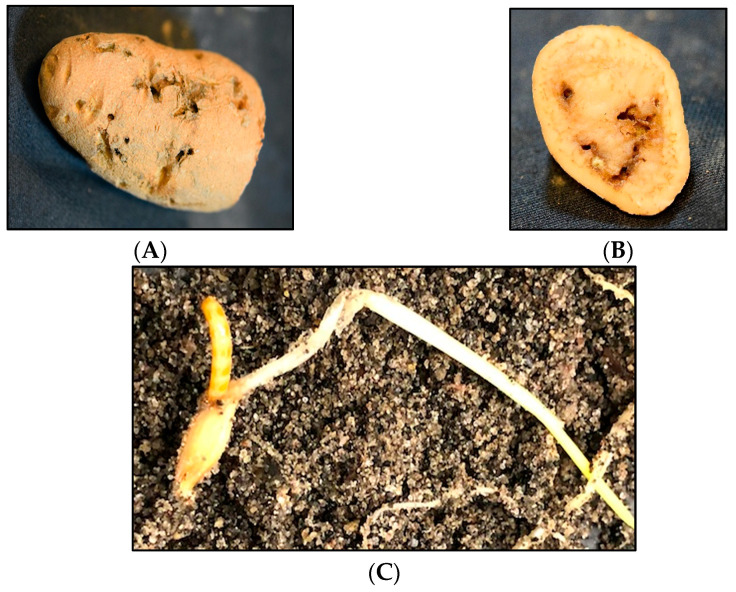
Feeding damage by the sugar beet wireworm, *Limonius californicus*. (**A**,**B**) Wireworm feeding damage in a potato, and (**C**) feeding on a barley seed.

## Data Availability

Not applicable.

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
