# Peer review of "Integrated Pest Management of Wireworms (Coleoptera: Elateridae) and the Rhizosphere in Agroecosystems"

_insects, 2022, doi:10.3390/insects13090769_

Round 1

Reviewer 1 Report

This manuscript is a mini-review type paper trying to introduce some of the existing challenges in wireworm IPM and discusses the potential impacts of various control methods on the rhizosphere.

 I found this manuscript in its present form wanting in several respects. It is not fully developed, neither concerning ipm nor concerning rhizosphere. And the link between these two parts is sometimes nil. It looks like the merging of two texts dealing with their subject in an incomplete way.

 Overall, it is not of the high standard expected from a review type manuscript published in a revue such MDPI Insects.

Author Response

Response to reviewer 1:

“I found this manuscript in its present form wanting in several respects. It is not fully developed, neither concerning IPM nor concerning rhizosphere. And the link between these two parts is sometimes nil. It looks like the merging of two texts dealing with their subject in an incomplete way.”

Response: We agreed and tried to address this shortfall to improve the presentation. We also believe that some of the missing connections are gaps in the existing knowledge, and we tried to highlight those areas in our revised version.

We organized this manuscript to highlight the impact of various IPM practices on wireworms and then discussed the effects of those practices on the rhizosphere and the soil microbiome; negative impacts on soil health parameters would have negative effects on the productivity of the agricultural system. The result of changes in the rhizosphere on wireworms is also introduced, where the information is available. We recognize and agree that there are missing connections in some places, primarily because it is a developing area of research, and there are gaps in the existing literature and knowledge. Our revised version highlights these current gaps and identifies areas needing further investigation. We believe this manuscript will help develop research questions and studies that will improve the effectiveness of IPM strategies against wireworms and other subterranean arthropods and contribute to the resilience of our production systems.

Reviewer 2 Report

This review article (insects-1788520) by Drs. Nikoukar and Rashed is well motivated, the structure is appropriate, and the manuscript is well written. The originality of this manuscript is that it goes beyond the simple exploration of literature, the description of events and the listing of different alternatives to chemical treatments for wireworms. It highlights many recent results and comments on their possible effects on soil rhizosphere. Overall, I enjoyed reading this manuscript. A few remarks have been below for authors to consider.

L60-2: I’d suggest adding here a reference Environmental Science and Pollution Research (2020) 27, 29867–29899 to support this statement. Recommended study provides a global systematic assessment of IPM-based alternatives to systemic pesticides for managing wireworms (among other pests) in four major field crops.

L277-86: Some of these statements are controversial, in my opinion. Recent research has provided a strong argument that switching from a conventional tillage system to a no-till maize production may not cause an increase of wireworm damage to maize (see Crop Protection 2021, 149, 105751), even though no-till conditions have been historically associated with increased wireworm damage risk. No effects of tillage were associated with wireworm densities and beetle captures. Please revise accordingly.

L283-4: The effects of no till farming practices on wireworm pests in maize fields are inconclusive, and current recommendations may unjustifiably limit grower options. See Crop Protection (2021) 149, 105751.  

Author Response

Response to reviewer 2:

“L60-2: I’d suggest adding here a reference Environmental Science and Pollution Research (2020) 27, 29867–29899 to support this statement. Recommended study provides a global systematic assessment of IPM-based alternatives to systemic pesticides for managing wireworms (among other pests) in four major field crops.”

Response: Done (now lines 100-101)

“L277-86: Some of these statements are controversial, in my opinion. Recent research has provided a strong argument that switching from a conventional tillage system to a no-till maize production may not cause an increase of wireworm damage to maize (see Crop Protection 2021, 149, 105751), even though no-till conditions have been historically associated with increased wireworm damage risk. No effects of tillage were associated with wireworm densities and beetle captures. Please revise accordingly.”

Response: Done (now lines 397-400).

“L283-4: The effects of no till farming practices on wireworm pests in maize fields are inconclusive, and current recommendations may unjustifiably limit grower options. See Crop Protection (2021) 149, 105751.”  

Response: We agree and believe that this is an area that needs species- and location-specific studies. This is now highlighted in this section (now lines 400-402).

Reviewer 3 Report

The presented manuscript has a certain interest in the agroecology of insects. However, there are some questions for the authors.
1. Introduction it is necessary to clearly formulate the goals and objectives of the manuscript.

2. It seems to me that it would be useful to add a section on the world fauna of Elateridae at the very beginning of the manuscript.

3. The conclusion contains little information. I ask you to completely redo it.

Author Response

Response to reviewer 3:

“Introduction it is necessary to clearly formulate the goals and objectives of the manuscript.”

Response: Done- The last paragraph of the introduction is now revised to clarify the objectives (now lines 126-132)

“It seems to me that it would be useful to add a section on the world fauna of Elateridae at the very beginning of the manuscript.”

Response: Done.

The conclusion contains little information. I ask you to completely redo it.”

Response: The conclusion is revised and expanded to also include examples of future directions.